# The Effect of Pixel Design and Operation Conditions on Linear Output Range of 4T CMOS Image Sensors

**DOI:** 10.3390/s24061841

**Published:** 2024-03-13

**Authors:** Wenxuan Zhang, Xing Xu, Zhengxi Cheng

**Affiliations:** Shanghai Institute of Technical Physics, Chinese Academy of Sciences, Shanghai 200083, China; zhangwenxuan@mail.sitp.ac.cn (W.Z.); xuxing@mail.sitp.ac.cn (X.X.)

**Keywords:** CMOS image sensor, 4-Transistor, linear output range, pixel design, operation conditions

## Abstract

We analyze several factors that affect the linear output range of CMOS image sensors, including charge transfer time, reset transistor supply voltage, the capacitance of integration capacitor, the n-well doping of the pinned photodiode (PPD) and the output buffer. The test chips are fabricated with 0.18 μm CMOS image sensor (CIS) process and comprise six channels. Channels B1 and B2 are 10 μm pixels and channels B3–B6 are 20 μm pixels, with corresponding pixel arrays of 1 × 2560 and 1 × 1280 respectively. The floating diffusion (FD) capacitance varies from 10 fF to 23.3 fF, and two different designs were employed for the n-well doping in PPD. The experimental results indicate that optimizing the FD capacitance and PPD design can enhance the linear output range by 37% and 32%, respectively. For larger pixel sizes, extending the transfer gate (TG) sampling time leads to an increase of over 60% in the linear output range. Furthermore, optimizing the design of the output buffer can alleviate restrictions on the linear output range. The lower reset voltage for noise reduction does not exhibit a significant impact on the linear output range. Furthermore, these methods can enhance the linear output range without significantly amplifying the readout noise. These findings indicate that the linear output range of pixels is not only influenced by pixel design but also by operational conditions. Finally, we conducted a detailed analysis of the impact of PPD n-well doping concentration and TG sampling time on the linear output range. This provides designers with a clear understanding of how nonlinearity is introduced into pixels, offering valuable insight in the design of highly linear pixels.

## 1. Introduction

Over the past few decades, the CMOS image sensor has gradually expanded its market share in the image sensor industry, becoming the mainstream image sensor technology and finding extensive applications in almost every field [1]. The linear response of an image sensor is characterized by the output signal changing linearly with the intensity of incident light. To better display the image, many commercial photography devices have a non-linear response to light intensity [2]. Non-linearity is intentionally implemented for purposes like high-dynamic-range image sensors [3,4], or to match human visual perception [5]. But for accurate measurements, like in some medical and scientific imaging fields, optical sensors require precise light intensity and frequency responses [6], where the linearity of the sensor becomes an important parameter. A lower level of nonlinearity can also enhance the accuracy of calculating parameters such as charge conversion gain or distance [7,8].

For typical 4-Transistor (4T) pixels, the PPD is the p+np structural photodiode utilized for photoelectric conversion. In this configuration, the nwell region of the PPD serves as the source of a transistor, with the Floating Diffusion (FD) acting as the drain. The FD is responsible for temporarily storing the signal electrons transferred from the PPD, as shown in Figure 1. When increasing the light intensity or exposure time, more photoelectrons will accumulate in the PPD; the electrons are converted to a voltage change in FD and result in a larger output signal. This is a rather linear process. As the PPD gradually reaches its full well capacity at a high light level, excessive photoelectrons will not be read as an effective output signal, and the pixel’s output voltage will tend to saturate. This degrades the linearity of the pixel. Several notable studies have previously investigated the sources of nonlinearity in 4T pixels. Through improvements in circuit structure and methods such as digital calibration, these works have successfully mitigated the nonlinear response, achieving a final output linearity exceeding 99.95% [9,10,11]. Within the linear region, nonlinearity is mainly caused by the variation in electrical parameters with voltage, such as the small-signal voltage gain of the source follower (SF) and the capacitance of the FD (C_FD_) node [12]. The nonlinearity of CFD can be mitigated by reducing the capacitance value, and the linearity range of SF can be enhanced through the use of isolated p-well and deep n-well structures [13,14]. In terms of circuit structure, replacing the SF with p-MOS or an analog buffer, along with the addition of a capacitive trans-impedance amplifier (CTIA), has been proposed to mitigate the nonlinearity of the SF and C_FD_. Replacing the SF cannot significantly reduce the nonlinearity. However, CTIA will introduce much more noise and increase the dark current [10]. Digital calibration involves neutralizing nonlinearity in the output through a nonlinear ramp voltage generator [5,10]. In these studies, the optimization methods can be divided into the analog domain and digital domain. Achieving an extremely high linearity is mostly realized through digital domain; however, digital calibration is a method that consumes significant chip space and power [9]. This underscores that the optimization at the existing hardware level is still insufficient for the design of high-linearity pixels.

In this study, our focus lies on enhancing the range of linear responses, and exploring more inherent parameters within the analog domain of 4T pixels. The essence of output nonlinearity lies in the variation in electrical parameters with output, and a higher nonlinearity inevitably leads to a reduction in the linear output range. Therefore, investigating the linear output range is a more straightforward and expedient approach.

This study centers on investigating the linear output range of 4T pixels and strategies for enhancing this range. The second section will provide a macroscopic analysis of factors influencing the linear output range. The third section will introduce the testing equipment and conditions, along with the data acquisition process. In the fourth and fifth sections, we will analyze the effects of pixel design and circuit control parameters, respectively, on the linear output range. In Section 6, we will examine the impact of parameter adjustments on noise and offer a more detailed discussion on the influence of PPD design and TG transfer time on the linear output range. Finally, Section 7 will summarize the results and analysis presented in this paper.

## 2. Influential Factors and Analysis

The structure of the 4T pixel and subsequent readout circuitry is illustrated in Figure 1; numerous factors regarding this structure may influence the linear output range of pixels. The progression from incident light to digital signal output can be divided into four distinct stages: photo-detection, electron-to-voltage conversion, voltage amplification, and the final analog-to-digital signal conversion [12]. Each of these stages has the potential to introduce nonlinearity. In this context, we posit that the processes of photo-detection and analog-to-digital conversion are ideal, with a specific focus on examining potential nonlinearity in the intermediate steps of electron-to-voltage conversion and voltage amplification. A portion of these influences arises from the CV characteristics of various circuit components, including the variations in C_FD_ and the gain in the SF with respect to output voltage, as well as the inherent linear output range of the circuit components, such as the output buffer and analog-to-digital converter. On the other hand, constraints on electron transfer efficiency, such as the design of TG sampling time and PPD doping concentration, also play a role. CV characteristics manifest in the output as nonlinearity due to variations in capacitance and gain, preventing the generation of an equivalent output voltage for the same amount of signal electrons. An insufficient charge transfer may result in the ineffective readout of signal electrons, consequently introducing nonlinearity. This phenomenon becomes more pronounced, particularly when more electrons accumulate in the PPD.

The linear output range is defined as the maximum voltage range at which the output linearity is ≥98%. This can be roughly determined using the following formula, wherein P represents the input light power, and V represents the corresponding output voltage. Four key factors contributing to nonlinearity have been identified: CG denotes the conversion gain of the FD capacitor, Q_TX_ is the transfer efficiency, and GSF and Gbuffer represent the small signal gain of the SF and output buffer, respectively. The denominators of each term are the values of these parameters at zero input. The numerators are the values at a certain light power. The independent variable of each parameter is labeled as P or V based on whether electron-to-voltage conversion has occurred. The overall nonlinearity effects of these factors are computed by multiplying them together to ascertain the total nonlinearity. To determine the linear output range, one must identify the corresponding P and V values where the nonlinearity calculated on the left-hand side is equal to 2%.
(1)$$2\%=\left|1-\frac{CG(P)}{CG(0)}\times \frac{Q_{TX}(P)}{Q_{TX}(0)}\times \frac{G_{SF}(V)}{G_{SF}(0)}\times \frac{G_{buffer}(V)}{G_{buffer}(0)}\right|$$

In this map, we investigate several parameters, marked in red circles, that may affect the pixel’s linear output range. These factors are categorized into two groups: pixel design and circuit control. Pixel design factors include the design of the PPD and the size of the FD capacitance. Circuit control examines the TG on-time, the reset voltage of FD, and the presence or absence of subsequent output buffers.

## 3. Experimental Setup and Data Process

The employed pixels are fabricated using the 0.18 μm 3.3 V CIS process. The pixel array consists of six channels, with B1 and B2 as two 10 μm pixel channels at a scale of 1 × 2560, while B3 to B6 are 20 μm channels at a scale of 1 × 1280. Each channel employs separate left and right readouts. Two different doping concentrations were employed for n-well in the PPD, namely standard doping (PPD-SD) and low-dose doping (PPD-LD). The full-well capacity of the PPD-LD is half that of PPD-SD. Figure 2a shows the graph of the CIS chip.

The schematic diagram of the PCB test board is presented in Figure 2b: control signals for the CIS chip are generated by an FPGA. Utilizing a correlated double-sampling (CDS) technique, the output signals of each pixel are read, and these two signals are subsequently input into a differential amplifier to minimize readout noise. The signals are acquired using the National Instruments PXIe-6386 Multifunction I/O Module, and a pre-developed Labview program is employed to read 100 frames of data.

During testing, the output of the left half of the pixels is read, with each reading comprising 100 frames on average. Figure 3 illustrates the output response curves of 340-pixel elements from each of the six channels, all under the same illumination intensity. Notably, the integration time for the 10 μm channels is half that of the 20 μm channels.

To better display the detail of the curves, the light power range in different figures is adjusted correspondingly using a light reduction film. The *x* axes are labeled as “relative light intensity”. There are differences in the absolute light intensity corresponding to x-coordinate in each figure. For reference, the scale of 10,000 in Figure 1 is approximately 2000 lux.

As illustrated in Figure 4, the output of the left 640 pixels from Channel B6 is presented, comprising data from 100 frames. This representation shows the typical output of pixels under non-saturation conditions. Due to packaging constraints, the light at the edges is partially shadowed, resulting in weaker responses from these pixels. To ensure statistical accuracy, we excluded 300 pixels near the edge, as documented in Table 1.

Each test is conducted under specific testing conditions, and if not specified, the general testing parameters are provided in Table 1:

**Table 1 sensors-24-01841-t001:** General testing conditions.

Integration Time	20 μm:280 μs; 10 μm:140 μs
TG Sampling Time	3.6 μs
Number of Test Frames	100 frames
Vdd Reset Voltage	3.3 V
Output buffer	1× output buffer
PPD type	PPD-SD
FD Capacitance	As shown in Figure 5
Pixel count	20 μm: Left No.301-640; 10 μm: Left No.941-1280

**Figure 5 sensors-24-01841-f005:**
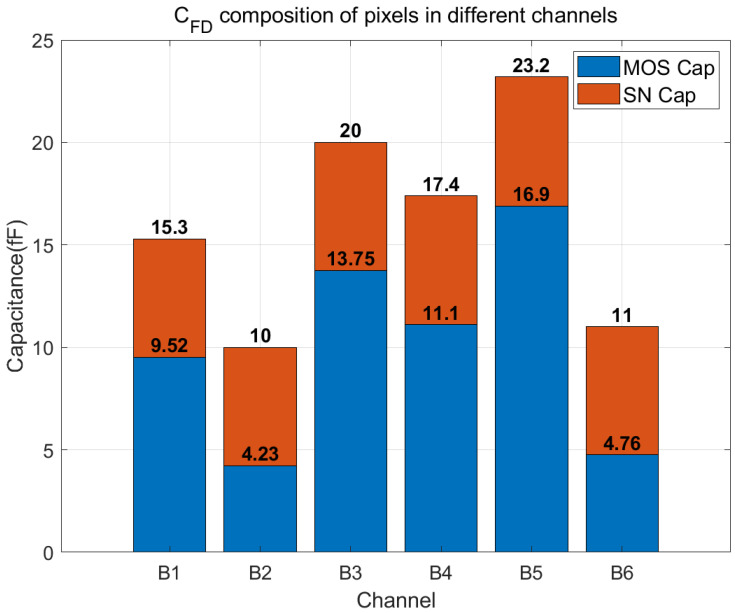
The components of C_FD_ for all six channels, with red bare representing the SN capacitance extracted from the layout and blue bar representing the additional MOS capacitance.

## 4. Influence of Pixel design

### 4.1. FD Capacitance

The FD capacitance encompasses the junction capacitance of the FD node and parasitic capacitance, collectively categorized as the Storage Node (SN) capacitance. Another portion is a deliberately incorporated MOS capacitor, connected in parallel with the FD node to adjust the overall C_FD_. The SN capacitance is approximately 6 fF across all channels, while the MOS capacitance varies for each channel, as illustrated in Figure 5. The conditions of each channel are summarized in Table 2.

Figure 6 shows how the linear output range changes for 10 μm and 20 μm channels with respect to C_FD_. It is evident from the graph that, for pixels of the same dimensions, there is a notable reduction in the linear output range with the increase in C_FD_. The linear output range of 20 μm pixels is larger than that of 10 μm pixels. This is attributed to the different TG structures of the two types of pixels: the TG of 20 μm pixels is centrally positioned, while the TG of 10 μm pixels is located on both sides. This positioning indicates that a centralized collection structure is advantageous for enhancing linearity.

Due to variations in the capacitance values of the pn junction and MOS capacitors with the applied bias, the FD capacitor in the 4T pixel introduces substantial nonlinearity. With the increase in output voltage, there is a corresponding alteration in the bias applied to C_FD_, leading to an elevation of the C_FD_ [12,13]. This results in a drop in charge conversion gain as the output voltage increases, causing a noticeable downward bend in the output curve. Notably, a larger FD capacitance amplifies this nonlinearity, leading to a reduction in the pixel’s linear output range. For 20 μm pixels, reducing C_FD_ from 23.2 fF to 10 fF results in a 37% increase in the linear output range. Similarly, for 10 μm pixels, decreasing C_FD_ from 15.3 fF to 10 fF also leads to a gain of over 30% in the linear output range.

### 4.2. PPD Design

In this section, we examine the impact of two distinct PPD designs on the pixel’s linear output range. PPD-SD serves as the reference standard design, and unless otherwise specified, this design is consistently employed. PPD-LD incorporates a lightly doped n-well, with the dose being half of that in PPD-SD. The full-well capacity of PPD-SD and PPD-LD are approximately 4.0 ke-/μm^2^ and 2.4 ke-/μm^2^, respectively.

Figure 7 reveals that, for various pixels with distinct PPD designs, their linear output range is inversely correlated with C_FD_. Notably, PPD-LD, subjected to n-well doping at a reduced dose, exhibits a markedly greater output linear range than PPD-SD across all channels. Among them, the enhancement is most significant in the B5 channel, where the linear output range increased by nearly 0.4 V, representing an improvement ratio of 32%. In contrast, the B6 channel, which already had a large linear output range, experienced an increase of less than 0.1 V in its linear output range. A detailed analysis of the linear output range with respect to PPD type will be provided in the discussion section.

## 5. Influence of Circuit Control

### 5.1. TG Time

Figure 8 illustrates the variation in the linear output range of channels B4–B6, concerning the TG sampling time (TG time), and the integration time is set to 560 μs. When the TG time is relatively long (>3.6 μs), the linear output range of pixels in each channel stabilizes at a fixed value. As the TG time decreases to below 3.6 μs, however, the linear output range of pixels swiftly diminishes. As TG time changes from 0.4 μs to 24 μs, the improvement in the linear output range can be as large as 60%. Attributed to the larger pixel size and the absence of specialized acceleration structures, this phenomenon leads to an inadequate electron transfer within the shorter timeframe.

Figure 9 depicts the output curve sets of 340 pixels for Channel B6 at TG times of 0.4 us, 1.6 us, and 7.2 us, respectively. The three sets of curves exhibit good alignment in the lower output voltage range. As the output signal increases, the inadequate charge transfer becomes evident. This is reflected in the smaller output signals for curve sets with shorter TG times under the same light intensity. Additionally, excessively short TG times amplify the non-uniformity during the transfer process, resulting in greater dispersion among curve sets with shorter TG times. Figure 9b provides an enlarged view of the three sets of curves, offering a clearer illustration that shorter TG times correspond to lower-positioned and more dispersed curve sets. Furthermore, as observed from the graph, the issue of transfer efficiency becomes more pronounced when there is a higher quantity of signal charge. This could potentially be one of the reasons why PPD-SD exhibits a lower linear output range compared to PPD-LD.

### 5.2. Supply Voltage of Reset Transistor

The Vdd_rst voltage is adjusted through a variable resistor, and this voltage is measured at the corresponding pin on the chip. Due to a slight drop in voltage, the measured value at this pin is approximately 3.1 V when a supply voltage of 3.3 V is applied. In a broad voltage range, the Vdd of the reset transistor has no significant impact on the linearity of pixel output, as shown in Figure 10. Reducing Vdd_rst proves beneficial for lowering readout noise [14], as the lower reset voltage diminishes the potential difference between the reset transistor’s source and drain and the surrounding environment. This enables designers to achieve a lower pixel noise through a soft resetting with a lower reset voltage. In the tested samples, the output swing is primarily constrained by the FD, and as long as Vdd_rst is not excessively decreased, it does not affect the output characteristics.

### 5.3. Output Buffer

In the design of the readout circuit, an output buffer was placed at the end of the readout chain to enhance the driving force of the signal. The buffer is a switched-capacitor amplifier with unity gain, as shown in Figure 11. The figure displays two sets of test results for Channel B1: with and without an output buffer. When there is no output buffer, the output voltage is negative, and these, are flipped in Figure 11. The results reveal that, in the absence of an output buffer, the saturated output voltage of the pixel increases by 10%, while the linear output range expands by 26%. It is evident that the presence of an output buffer somewhat restricts the linear output range of the pixel.

This indicates that consequent circuit structures like the output buffer, which enhance the driving capabilities, should be carefully designed or this will become one of the major limitations of the linear output range.

## 6. Discussion

Based on previous data and analysis, it is evident that methods such as reducing the PPD n-well doping concentration, adjusting the C_FD_, and extending the TG time within a certain range can significantly enhance the linear output range of pixels. In this section, we will investigate whether these improvements introduce additional readout noise and delve deeper into the mechanisms behind the optimization of n-well doping and TG time. Additional noise canceling techniques can also be implemented [15,16].

The following figure illustrates the magnitude of noise voltage for each channel under different conditions, with Fixed Pattern Noise (FPN) contributing the most to the overall output noise. In our statistical analysis, we excluded the FPN and calculated the standard deviation of the remaining data over a 100-frame period, and then averaged the noise data for 340-pixel elements. Figure 12a displays the noise data for several channel pixels under varying TG times, while Figure 12b compares the noise differences between PPD-SD and PPD-LD.

For the B6 channel pixels with a smaller FD capacitance, the output noise voltage increases with longer TG times, but for channels with a larger FD capacitance, extending TG time does not result in significant changes in noise. It is noteworthy that, based on the test results in Section 5.1, extending the TG time beyond several microseconds saturates the linear output range of pixels, and adapting a longer TG time is actually meaningless. The results in Figure 12b demonstrate that lower-concentration n-well doping has no discernible impact on noise, and the influence of FD size on noise can be disregarded. This indicates that by optimizing factors such as the FD capacitor, PPD n-well doping, and TG time, it is possible to significantly enhance the linear output range of pixels without introducing a noticeable impact on output noise.

To understand the reason for the phenomenon wherein a pixel characterized by a diminished PPD capacitance exhibits an expanded linear output range, an examination of the charge transfer process is imperative. Figure 13a presents potential diagrams illustrating the trajectory of electron transfer, starting from the edge of the PPD, passing through the channel beneath TG, and concluding at the FD node. During stage A, when the number of photoelectrons within the PPD is relatively low, the activation of TG results in the transfer of photoelectrons to the FD node, inducing a discernible drop in voltage across the FD. Progressing from stage A to B, the count of photoelectrons remains sufficiently modest, and the FD is still sufficient to accommodate all photoelectrons. Ignoring the variations in C_FD_ with respect to the bias voltage, as mentioned earlier, the voltage drop across the FD node steadily escalates in a linear fashion with the augmentation in the count of photoelectrons.

However, a crucial consideration arises due to the presence of a pinning potential (Vpin) within the PPD, compounded by the discrepancy in capacitance between PPD and FD. When there are excessive photoelectrons, a redistribution occurs between the two capacitors through charge-sharing, as illustrated in Figure 13a. Hence, when the voltage on the FD node attains the critical value Vpin, an equivalent voltage drop can no longer be sustained by the same number of electrons, causing the output curve’s departure from the linear region. The process-induced deviation of Vpin also causes non-uniformity among the families of curves. Notably, the pinning voltage of PPD-LD is significantly diminished owing to the reduced n-well doping.

Furthermore, it can be observed that PPD-LD exhibits a more pronounced improvement in channels with a larger FD. This is attributed to the fact that, under the given test conditions of integration time and TG time, around 1.6 V represents the upper limit of the linear output range that is achievable using the pixels. This also explains the weaker correlation between the linear output range and C_FD_ in the data for PPD-LD channels.

In the analysis presented in Section 2, we posited that the linear output range is directly linked to the charge transfer efficiency, Q_TX_. Consequently, it is necessary to scrutinize the charge transfer process and, based on this, develop a parameter model to fit the variation in the linear output range with respect to TG time. Figure 14 depicts a one-dimensional PPD model with a total length L. Given the larger PPD dimensions considered in this study, the charge transfer process is predominantly governed by the diffusion mechanism [17].

During the transfer process, electrons near the TG are rapidly swept into the FD node through the drift process. As the drift process is significantly faster than the diffusion process, the calculation of the total electron transfer time only needs to consider the diffusion process. Simultaneously, the influence range of the electric field near the TG and at the edge of the PPD is limited, so the length over which the farthest electrons need to diffuse is approximately L. Considering a PPD irradiated with a certain amount of light after a certain time, as illustrated in Figure 15, N(t,x) represents the electron line density at a specific position in the PPD at a given moment, and N(0) represents the maximum charge line density in the PPD at the initial moment.

Initially, the electron distribution function near the TG has an infinite slope, leading to the rapid diffusion of some electrons. After the diffusion of some electrons, the slope of the N distribution (i.e., the gradient of electron density) decreases, and the diffusion process becomes progressively slower. Mathematically, as the gradient of the final charge density tends toward zero, Q_TX_ does not reach 100%, and it is only possible to calculate the transfer time of a specific Q_TX_ value. For ease of calculation, electron diffusion is divided into two processes: the first process transfers half of the charge, and the second process transfers the remaining portion of the charge, as illustrated in Figure 15. At thermal equilibrium, the diffusion equation is expressed as follows:(2)$$J_{n}=qD\frac{\partial N}{\partial x}$$
where J_n_ represents the diffusion current density, and D is the corresponding diffusion coefficient. In the first stage, the equation can be formulated based on charge conservation as follows:(3)$$q\frac{l}{2}N(0)=\left|\int_{0}^{t}-qD\frac{N(0)}{l(t)}dt\right|$$

The left-hand side of the equation represents the total number of electrons diffusing during time t, which can also be calculated using the expression on the right-hand side through the distribution function N, as illustrated in Figure 15. Accordingly, we have:(4)$$l(t)=2\sqrt{Dt}$$
Let l = L, we can yield the transfer time of the first stage as follows:(5)$$T_{1}=\frac{L^{2}}{4D}$$
Similarly, using charge conservation for the second stage, it can be expressed as:(6)$$q\frac{l}{2}N(t)=q\frac{L}{2}N(0)-\left|\int_{T_{1}}^{t}-qD_{n}\frac{N(t)}{L}dt\right|$$
If the remaining charge proportion is denoted as p, and:(7)$$p=1-Q_{TX}(p<0.5)$$
The time for the second stage T_2_ can be solved when the remaining charge proportion is p:(8)$$J_{n}=qD\frac{\partial N}{\partial x}T_{2}=-\frac{L^{2}}{2D}ln(2p)$$

The above analysis establishes a physics-based relationship between charge transfer time and transfer efficiency. Assuming the linear output range is proportional to Q_TX_ with a coefficient C, we can roughly construct a mathematical relationship between TG time and the linear output range. The constant part can be set as unknown parameters. By then using the constructed parameter expression to fit the data points, we can derive a curve depicting the relationship between the linear output range and TG time:(9)$$T_{Total}=\frac{L^{2}}{4D}-\frac{L^{2}}{2D}ln(2p)=A-Bln(2-2Q_{TX})$$
(10)$$\mathrm{Linear}\mathrm{Output}\mathrm{Range}=C\left(1-exp\left(\frac{A-\mathrm{TG}\mathrm{Time}}{B}\right)\right)$$

The fitting curve in Figure 8 is based on this expression, demonstrating that when TG time is relatively short, extending TG time leads to a rapid increase in the linear output range, and as TG time becomes longer, the linear output range tends to saturate.

## 7. Conclusions

In this study, we examined several factors that constrain the linear output range of 4-T pixels. Based on the test results, we conclude that, to achieve a higher linear output range, it is advisable to limit the capacitance of the FD node and the pinning voltage of the PPD at the pixel design level. From a circuit control perspective, a too-short TG time severely degrades the linear output range of large-sized pixels. The subsequent circuitry’s buffer also emerges as a critical limiting factor in circuit design. Notably, variations in the reset voltage of FD within a significant range do not affect the output linearity. These findings offer valuable insights for the design of highly linear pixels and may even inspire certain pixel designs that intentionally introduce nonlinearity.

## Figures and Tables

**Figure 1 sensors-24-01841-f001:**
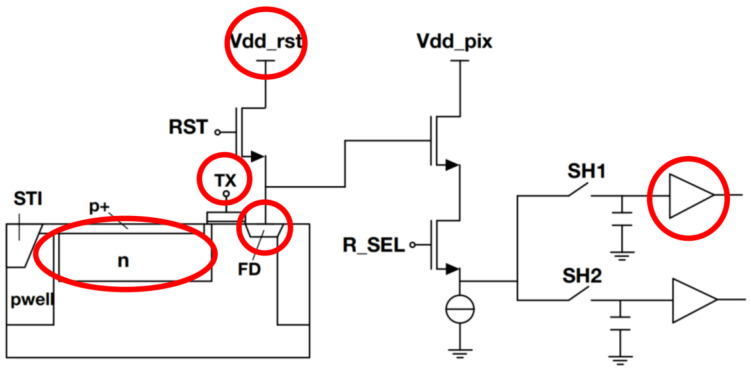
An illustrative structure for a 4T pixel and its read-out circuit. The structures and circuit parameters investigated in this work are marked with a red circle.

**Figure 2 sensors-24-01841-f002:**
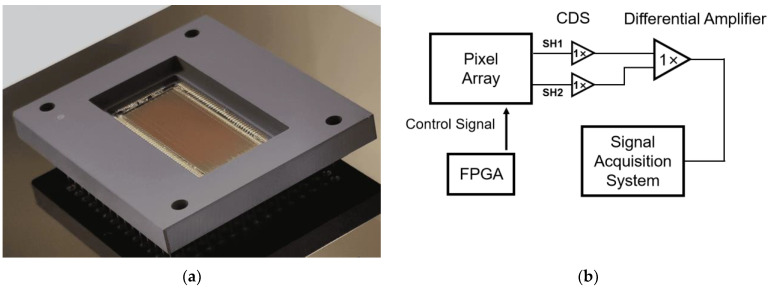
(**a**) Photo of the test chip; (**b**) the basic diagram of the PCB, illustrating the fundamental structure of the test circuit.

**Figure 3 sensors-24-01841-f003:**
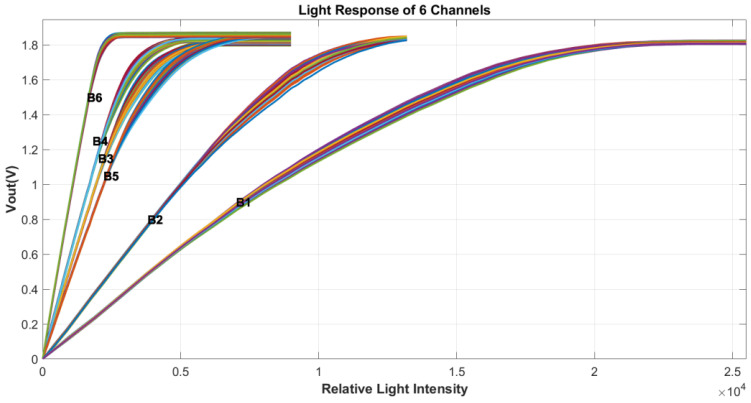
The output of all six channels under the same illumination conditions. The cluster of curves for each channel contains data for 340 pixels.

**Figure 4 sensors-24-01841-f004:**
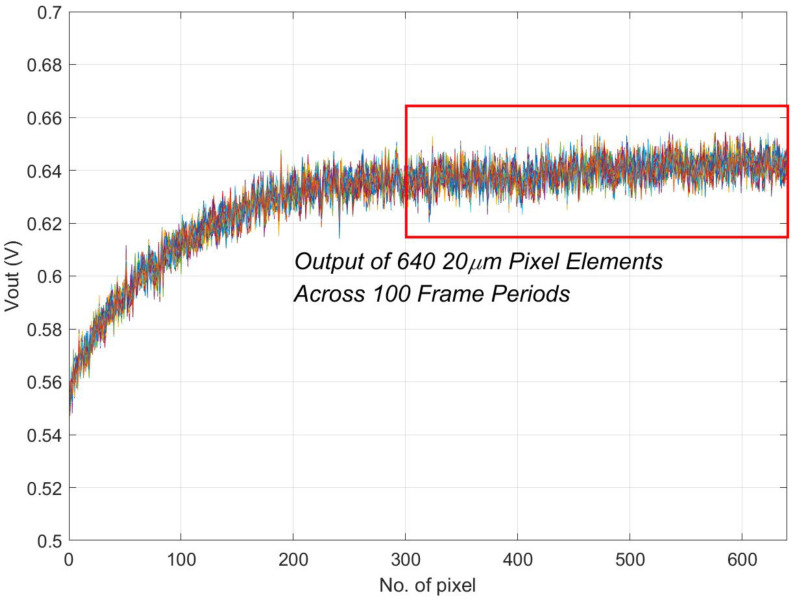
Output of 100 frames from the left 640 pixels of a 20 μm pixel channel. At this point, to prevent saturation, the input light power that is adapted is relatively low. Pixels near the edge exhibit smaller output signals due to shadowing.

**Figure 6 sensors-24-01841-f006:**
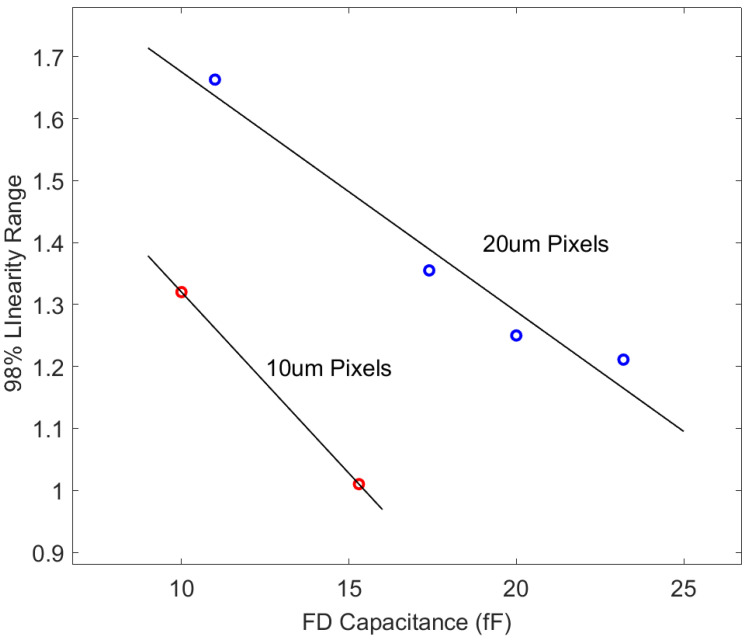
Relationship between C_FD_ and corresponding linear output range of each channel. Data are fitted for 10 μm and 20 μm pixels.

**Figure 7 sensors-24-01841-f007:**
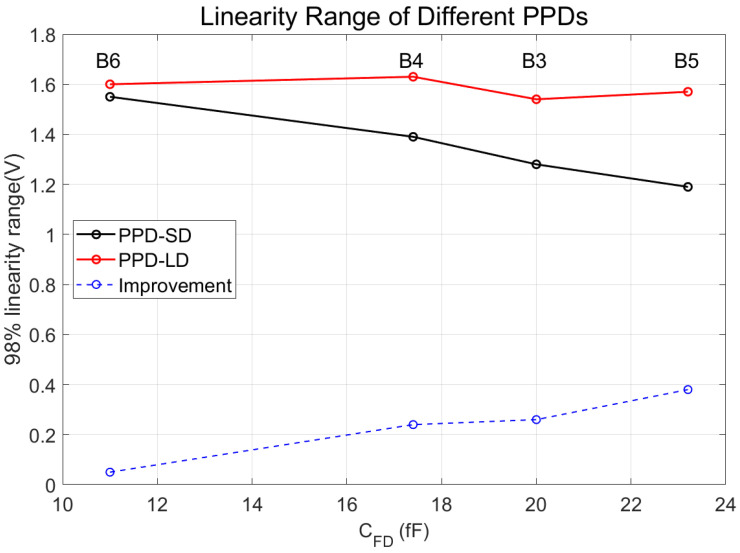
The linear output range of B3 to B6 channel with two types of PPD design with respect to C_FD_ (black line: PPD-SD; red line: PPD-LD). The blue dashed line is the increment in the linear output range for each channel.

**Figure 8 sensors-24-01841-f008:**
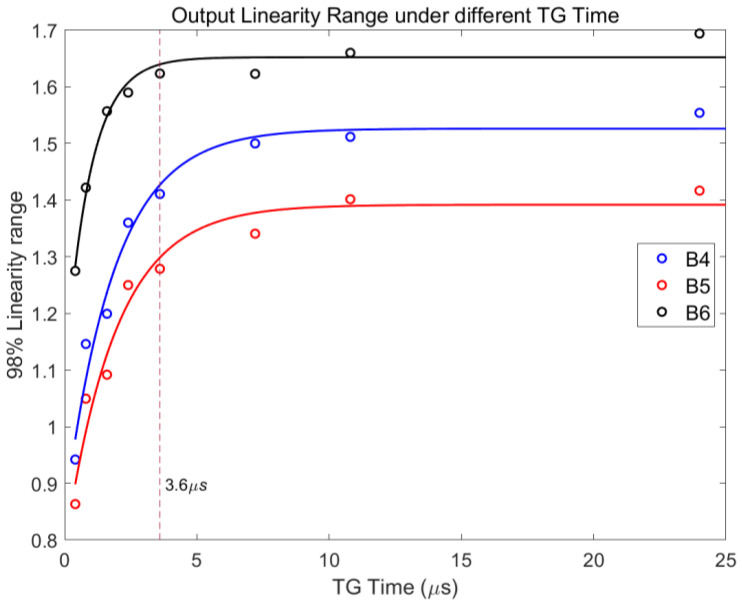
Relationship between linear output range and TG sampling time for channels B4–B6. The C_FD_ values of the three channels show an approximately arithmetic progression.

**Figure 9 sensors-24-01841-f009:**
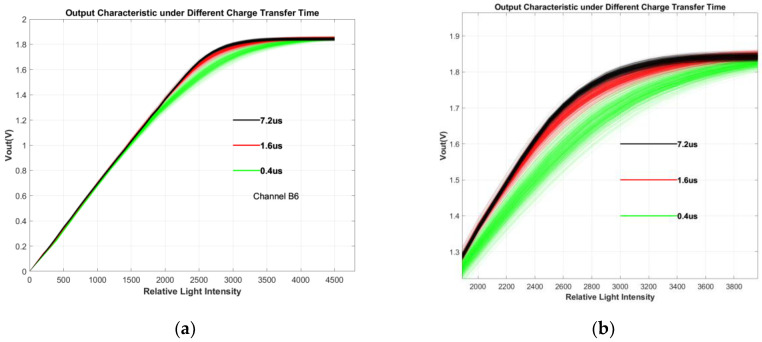
(**a**) The output curve of 340 pixels for channel B6 under 3 different TG times: 0.4 μs, 1.6 μs and 7.2 μs respectively; (**b**) partial magnification of (**a**), providing a clearer illustration of the curves from the linear region to the nonlinear region.

**Figure 10 sensors-24-01841-f010:**
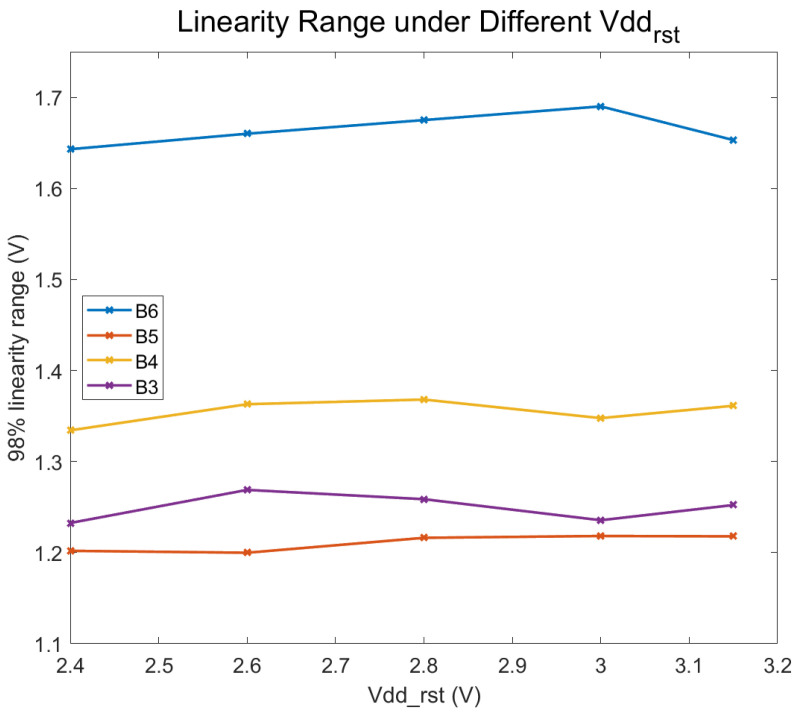
Relationship between pixels linear output range and the Vdd of reset transistor.

**Figure 11 sensors-24-01841-f011:**
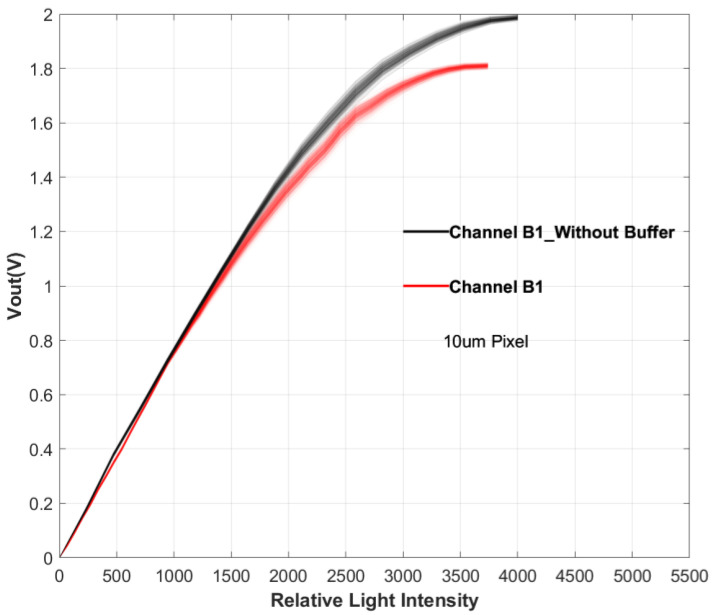
Comparison of transfer curves for 10 μm pixels with and without an output buffer.

**Figure 12 sensors-24-01841-f012:**
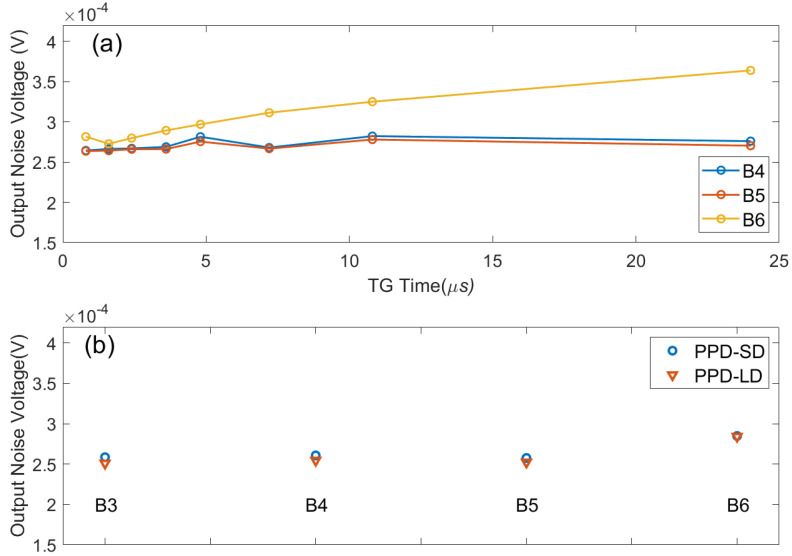
The read-out noise of 100 frames, averaged for 340 pixels under different conditions. (**a**) Readout noise for various TG times; (**b**) readout noise for PPD-SD and PPD-LD across four channels.

**Figure 13 sensors-24-01841-f013:**
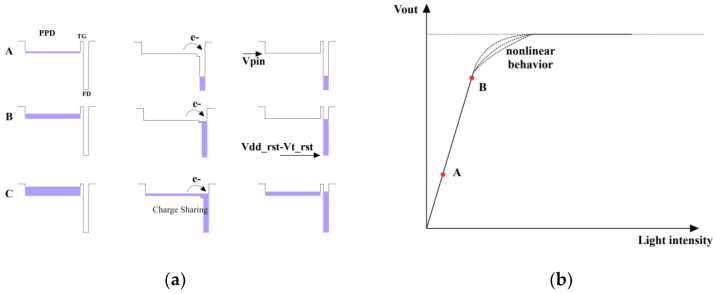
(**a**) The potential diagram of PPD-TG-FD structure during transfer process, where purple color represents the quantity of electrons within each area; (**b**) the style of output curve based on the potential diagrams given in (**a**).

**Figure 14 sensors-24-01841-f014:**
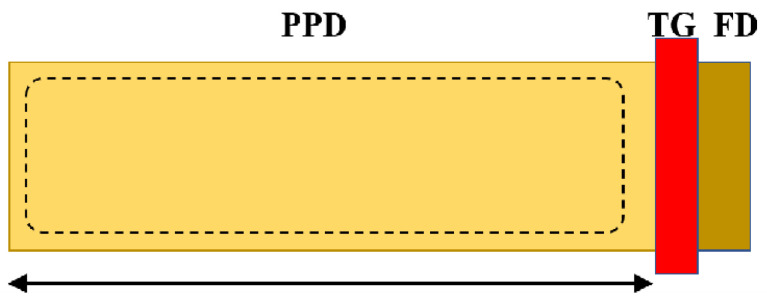
1D model of a large-sized PPD for calculating charge transfer time.

**Figure 15 sensors-24-01841-f015:**
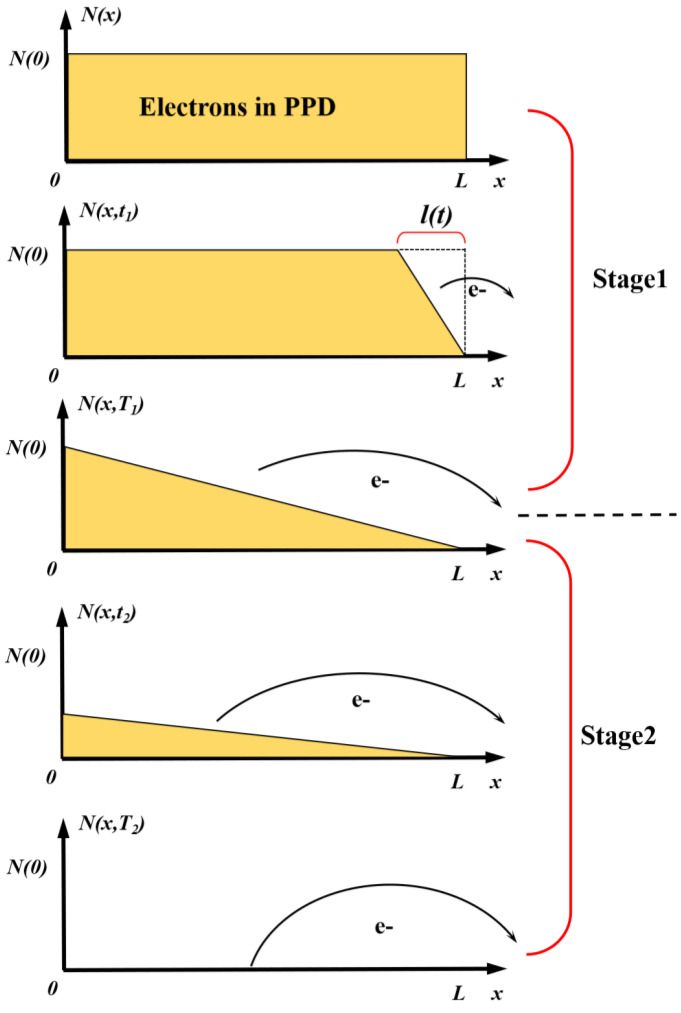
Mathematical model of the charge transfer process. Electrons gradually diffuse from the region near the TG edge to the collection node. For ease of calculation, the diffusion process is divided into two stages: the first stage transfers half of the electrons in the PPD, and the second stage transfers the remaining half.

**Table 2 sensors-24-01841-t002:** Conditions of the six channels.

Channel No.	C_FD_ (fF)	Pixel Size
B1	15.3	10 μm × 10 μm
B2	10.0	
B3	20.0	20 μm × 20 μm
B4	17.4	
B5	23.2	
B6	11.0	

## Data Availability

Data are contained within the article.

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
