# Peer review of "The Effect of Pixel Design and Operation Conditions on Linear Output Range of 4T CMOS Image Sensors"

_sensors, 2024, doi:10.3390/s24061841_

Round 1

Reviewer 1 Report

Comments and Suggestions for Authors

1. What is the capacitance of PD for each channel?

2. The reference numbers in the text are mixed up. It is best to number references in the order they appear in the text.

3. In the content of the paper, conjunctions such as but, yet, etc. were used like adverbs. It is necessary to check the overall paper and make corrections.

4. It is necessary to fully explain abbreviations such as PPD in the text.

5. There is no conjunction in the sentence on line 85, so you need to check it.

6. The content to be covered in this paper is summarized in Figure 1, but why was Source Follower not the subject? Is there a special reason? Even though it is a factor that greatly affects linear output, it is not mentioned.

7. It would be good to use a table to organize the conditions and characteristics of channels B1 to B6 at once.

8. Are the 3.6us and 3.1V presented in Table 1 appropriate conditions? It is necessary to mention the reasons for making that choice.

9. It seems necessary to mention the reason for the differences by channel in Figure 4.

10. I think there should be a graph of 3.6us in Figure 9.

Comments on the Quality of English Language

Please refer to my comments.

Author Response

Dear reviewer,

Thank you very much for taking time to read our manuscript and providing some extremely helpful suggestions. We also appreciate the detailed and specific listing of these suggestions. We have carefully reviewed each of your revision comments and spent several days making modifications and enhancements accordingly. After the revisions, the readability of the paper and the completeness of the data have indeed improved. Once again, we appreciate your efforts, and a detailed response to each of your revision comments is as follow:

  1. What is the capacitance of PD for each channel?

In section 4.2 of the revised manuscript, we have added relevant data on the full well capacity of both PPD-SD and PPD-LD to enhance the completeness of the dataset in the paper. The full well capacity of PPD-SD and PPD-LD are approximately 4.0 ke-/μm2 and 2.4ke-/μm2, respectively.

  1. The reference numbers in the text are mixed up. It is best to number references in the order they appear in the text.

The references are re-arranged based on the order in the text.

  1. In the content of the paper, conjunctions such as but, yet, etc. were used like adverbs. It is necessary to check the overall paper and make corrections.

We have thoroughly checked the entire manuscript, addressing several conjunction usage issues, such as 'Yet' and 'However'. They no longer function as adverbs by appearing at the beginning of  a sentence but rather effectively connect two clauses. Additionally, we have examined the entire text, rectifying minor grammatical errors and optimizing the use of several words. These changes have been underscored in the revised manuscript.

  1. It is necessary to fully explain abbreviations such as PPD in the text.

A more detailed explanation of some abbreviations is added in the Introduction section.

  1. There is no conjunction in the sentence on line 85, so you need to check it.

On the manuscript we originally uploaded, line 85 was the subtitle for the second section, which named Theory Analysis. The title of this section can indeed benefit from the addition of a conjunction and some improvement in words. We have modified it from 'Theory Analysis' to 'Influential Factors and Analysis. Other than that, we have thoroughly checked our use of conjunctions like mentioned in bullet point 3.

  1. The content to be covered in this paper is summarized in Figure 1, but why was Source Follower not the subject? Is there a special reason? Even though it is a factor that greatly affects linear output, it is not mentioned.

Thank you for pointing this out, it is indeed a limitation in our work. During the chip design process, our primary focus was on SF's noise characteristics. As a result, the design for SF was fixed in all channels and without corresponding test structures, making it challenging to directly measure GSF. Additionally, based on simulation results, the SF in our 4T pixel exhibits good linearity within the voltage swing, with GSF varying approximately from 0.912 to 0.917. This performance is attributed to specific process upgrades and optimizations undertaken by the Fab to address SF-related issues. In contrast, the corresponding literature reports lower linearity for SF (Fei Wang et al. 2017, Linearity Analysis of a CMOS Image Sensor[11]), with GSF around 0.85. Due to the relatively ideal performance of SF in our devices and the absence of specific test structures, we did not thoroughly investigate the impact of SF in our work.

  1. It would be good to use a table to organize the conditions and characteristics of channels B1 to B6 at once.

This is a good suggestion. We have added a table in Section 4.1 providing an overview of channels B1 to B6 to enhance the overall readability of the manuscript. The additional table is placed after Figure 5, which describes the capacitance components of each channel.

  1. Are the 3.6us and 3.1V presented in Table 1 appropriate conditions? It is necessary to mention the reasons for making that choice.

The choice of 3.6 us is due to its proximity at the turning point in the linear output range--TG time curve. It is also close to the typical value used during our testing of similar chips. The goal of Section 5.2 is also to investigate how to set the TG time more appropriately. For the study of other parameters, as long as TG time remains constant, it does not affect the corresponding conclusions.

As for the 3.1V reset voltage, the standard Vdd for this 0.18μm CIS process is 3.3V. 3.3V is also the actual voltage applied. The Vdd_rst voltage is adjusted through a variable resistor, and this voltage is measured at the corresponding pin on the chip. Due to a slight voltage drop, the measured value at this pin is approximately 3.1V when a supply voltage of 3.3V is applied. As we can only regulate Vdd_rst based on this measured value, the value used in Figure 10 is the measured one. For consistency with this figure, we wrote 3.1V when introducing the testing conditions. We have made this clear in the revised manuscript.

  1. It seems necessary to mention the reason for the differences by channel in Figure 4.

The reason of the differences by channels is mentioned in the paragraph before Figure 4. It is just because the packing is not perfect, and a diffused light source was used. The pixels near the edges are partially shadowed by the sidewalls, as shown in Figure 2a.

  1. I think there should be a graph of 3.6us in Figure 9.

The selection of 0.4μs, 1.6μs, and 7.2μs is intended for a clearer representation of the curve's form and the overall distribution of the curve family. Although 3.6 μs represents a typical testing condition, the family curve at this TG time overlaps too much with the family curve at longer TG times (7.2μs or longer). While increasing the transparency of the curves could address this issue, it results in overly blurred edges of the curve distribution, resulting in poor visual display. Through our experimentation, we found that the data at 1.6μs exhibits fewer overlaps with the preceding and succeeding curve families at turning point, providing the clearest representation of the family curve distribution and the changes with an extended TG time.

Reviewer 2 Report

Comments and Suggestions for Authors

The authors mention a detailed analysis of the impact of PPD n-well doping concentration on the linear output range, which needs to be more convincing. No values of n-well doping concentration are mentioned throughout the manuscript. Simply saying, sometimes, without values to support judgment, is meaningless.

This manuscript correlates the number of electrons and the linear output range. How are the electrons being quantified? The same goes for charge transfer efficiency. How does this be equalized with the linear output range performance?

Author Response

Dear reviewer, 

Thank you for pointing out various deficiencies in our work and our manuscript. Nothing is more conducive to improvement than some sharp criticisms. The author must acknowledge that writing is his biggest weakness in English learning, and this was evident in the IELTS test taken several years ago, where writing was his lowest band score subject. Although we have checked and revised the a number of grammar issues, several conjunction usage issues and the use some words, we still hope you can provide more specific criticism regarding the English expression of this paper, including vocabulary, grammar, and sentence structure etc. As we lack an English-speaking environment, receiving feedback of these is extremely valuable for us. We hope to hear from you again or receive further comments. The following is our response to all your comments and suggestions.

(1  As for the n-well concentration, another reviewer also mentioned a similar issue. Due to potential conflicts of interest, we are unable to access precise n-well doping conditions such as dose and energy. But indeed, we should provide some other relevant data. In the revised manuscript, we have included data on the full well capacity of two types of PPD (2.4ke-/μm2 for PPD-LD and 4.0ke-/μm2), see in section 4.2. Hope this will address your concern and improve the integrity of this work.

(2  About the quantification of the number of electrons and transfer efficiency: In the Discussion Section, the number of electrons is merely an intermediate result, resulting from pure mathematical processing. Our primary focus is on the ratio of electrons transferred and those retained. Transfer efficiency represents this ratio and is also a mathematical representation. What we want is a model that explains the relationship between TG time and linear output range: 1. The Discussion Section analyzes the relationship between charge transfer time and transfer efficiency in a large-sized PPD, where the charge transfer time can be directly linked to our TG sampling time; 2. In the analysis of the second section, we highlighted that nonlinearity is primarily influenced by FD capacitance, CV characteristics of several circuit components, and transfer efficiency. We assumed that the overall nonlinearity could be expressed as the product of these factors, i.e. linearity is directly correlated with transfer efficiency. When all other parameters are at their ideal values, considering the transfer efficiency itself introduces a 2% nonlinearity, the 98% linearity point can be determined. This allows us to establish a relationship between TG time and the linear output range. It should be noted that this is not a highly precise model; based on a physics-driven analysis, it provides an expression form, leaving some parameters undetermined, which need to be determined through fitting with data. Nevertheless, it can still explain the trend of the curves.

Kind Regards,

Wenxuan Zhang

Reviewer 3 Report

Comments and Suggestions for Authors

The authors present a systematic study on exploring the factors that affect the linear output of a 4-transistor CMOS image sensors. The results obtained is within reasonable assumption, based on which they experimentally demonstrate the connections between the linear response and the parameters such as the capacitance of the FD node, the pinning voltage of the PPD, and TG time, etc. The overall presentation of the manuscript is good, and the discussions are sufficient. Therefore, I would like to suggest its publication in the sensors.

Author Response

Dear reviewer,

Thank you very much for your recognition of our work. We would be delighted if our work could provide some value to researchers in the relevant field. Your approval is significant to us and serves as a crucial motivation, especially for the young researchers within our group. At the same time, we acknowledge that there are areas in our paper that could be improved.

Based on other reviewers' comments, we have addressed some grammar issues in the manuscript. We also add relevant data on pinned photodiode capacitance to enhance completeness and credibility of the results, including a table listing conditions of the six channels to improve readability. We also made A few clarifications in the manuscript regarding other reviewers concerns.

Thank you again for taking the time to read our paper and providing encouraging feedback.

Kind Regards,

The Authours